# NPR1-like genes in *Theobroma cacao*: Evolutionary insights and potential in enhancing resistance to *Phytophthora megakarya*

**Muhammad Umar Rasheed**[1], **Aiman Malik**[1], **Muhammad Zeshan Haider**[2], **Adnan Sami**[2], **Muhammad Shafiq**[1], **Qurban Ali**[1], **Muhammad Arshad Javed**[1], **Ansar Ali**[2,3]*

**1** Department of Plant Breeding and Genetics, Faculty of Agricultural Sciences, University of the Punjab, Lahore, Pakistan, **2** Graduate Institute of Biotechnology, National Chung Hsing University, and Academia Sinica, Taipei, Taiwan, **3** Department of Molecular Bioscience, University of Texas at Austin, Austin, Texas, United States of America

* ansarali@utexas.edu

## Abstract

Nonexpressor of pathogenesis-related 1 (NPR1) is crucial for activating the plant immune system through the signaling molecule salicylic acid (SA), which triggers systemic acquired resistance (SAR) in *Arabidopsis*. In this study, three putative genes associated with NPR1 from *Arabidopsis* have been identified in the genome of *Theobroma cacao*, namely, *TcNPR1, TcNPR2*, and *TcNPR3*, suggesting a functional diversification among the three gene entities. Phylogenetic analysis revealed that *TcNPR1* and *TcNPR2* branched alongside their Arabidopsis orthologs, *NPR1* and *NPR2*, indicating that these genes maintain a conserved role in SA signaling pathways across different species. In contrast, *TcNPR3* exists in a separate clade, suggesting unique functional roles and evolutionary divergence. A comparative analysis of the physiochemical properties of these *TcNPRs* showed a different subcellular localization, as *TcNPR1* persists in the cytoplasm, while *TcNPR3* is found in the nucleus, aligning with its proposed role in SA signaling and transcriptional regulation. Furthermore, we identified microRNAs that target *TcNPR3*, suggesting that *P. megakarya* may exploit the transcriptional regulatory network to bypass plant defense activation. Transient overexpression or suppression of *TcNPR* gene expression through RNA interference-mediated gene silencing could be sufficient to study the impact on the production of other molecules, such as SA, some PR protein expressions, and resistance against *P. megakarya*. The interactions between proteins encoded by *TcNPRs* and cellular proteins of *P. megakarya* will provide insight into whether the pathogen manipulates host defenses. Finally, the expression of *TcNPR* genes in response to infection by *P. megakarya* offers valuable information regarding the temporal and spatial activation during the defense response.

**Data availability statement:** All the relevant data is available within manuscript or at the following repository: https://www.kaggle.com/datasets/phytodoctor/npr1-dataset.

**Funding:** The author(s) received no specific funding for this work.

**Competing interests:** The authors declare no competing interest.

## Introduction

The Nonexpressor of pathogenesis-related 1 (*NPR1*) gene plays a vital role in systemic acquired resistance (SAR) mediated through salicylic acid (SA) in plants due to controlling immune responses through the modulation of pathogenesis-related (PR) gene expression [1]. Plants produce SA during a pathogen attack that may lead to signaling and then, within plasmodesmata, triggering the PR gene to hamper the spread of pathogens within cells. Increased expression of the PR gene up-regulates SA production, leading to programmed cell death at high concentrations; hence, a primary first line of defense is pathogen-associated molecular pattern-triggered immunity [2]. Recent investigations have demonstrated that NPR1 plays a vital role in activating the SAR, which in turn systemically migrates the signals from infected tissues to healthy ones and activates PR genes via SA accumulation [3,4]. A defense reaction of this type, often exhibited during this so-called priming period, is accelerated or further increased following subsequent exposure of plants to pathogenic attackers [5].

The NPR1 interacts with TGA transcription factors such as TGA2, TGA5, and TGA6 that bind to SA-responsive promoter elements for the turn-on of PR gene expression. In non-induced cells, the NPR1 oligomers remain inactive in the cytoplasm. Upon binding of SA, the oligomers dissociate, and NPR1 enters the nucleus to engage in transcriptional activation. The NPR1 protein contains a BTB/POZ domain, ankyrin repeats, and a C-terminal region considered necessary for its regulatory function. Although NPR1 was identified as a master regulator of SAR several decades ago, only recently has the direct interaction of NPR1 with SA been defined. The study also indicated that the NPR1 binds SA and synthetic SAR inducers such as BTH through Cys521/529 in a metal-dependent manner, inducing conformational changes and enhancing its transcription activity [6,7].

*T. cacao*, commonly known as "Food of the Gods," originates from the rainforests of tropical America, though it is of great economic importance in West Africa, Asia, and America [8,9]. West Africa produces about 71% of world cocoa production, supplying 4.3 million tons annually [10]. Despite the potential economic value, cocoa farming in such regions as Pakistan faces several challenges, including inadequately trained farmers and poor management of pests and diseases [11]. Its plant components, mainly concentrated in the bark at about 80% and seeds at 20%, undergo processing to obtain chocolate, a potential source of antioxidants, polyphenols, and dietary fiber. Cocoa plants are most susceptible to viruses such as Cocoa mild mosaic virus (CMMV), Cocoa Yellow Mosaic Virus (CYMV), and Cocoa swollen shoot virus (CSSV), which seriously affect the yield and quality of the beans [12]. Among the fungal pathogens, *P. megakarya* was identified as the significant CSSD contributor that resulted in drastic reductions in yield—from 4056 thousand tons in the year 2020 down to 3672 thousand tons in 2023 [13]. Based on recent findings, the participation of NPR1 in cacao immunity against *P. megakarya* has been underlined. Understanding NPR1 functionality in cacao may provide the key to understanding how resistance mechanisms could be improved and production losses from pathogen attacks mitigated. These findings emphasize the potential of molecular insights to inform global challenges in the cultivation and productivity of cocoa.

## Materials and methods

### Identification of *NPR1* gene in *T. cacao*

The amino acid sequence of *T. cacao* was obtained from the Cacao Genome Hub, an online database (https://cocoa-genome-hub.southgreen.fr/node/4), and the particular peptide sequence of *Arabidopsis thaliana* NPR-1 (NP_176610) was retrieved using a database of NCBI (https://www.ncbi.nlm.nih.gov/). NPR-1 (NP_176610) peptide sequences were used to identify NPR1-like genes in *T. cacao* (https://cocoa-genome-hub.southgreen.fr/node/4)

peptide using BLAST-p (Protein Basic Local Alignment Search Tool) in TBtool (an offline tool may be executed on any machine that supports the Java Runtime Environment 1.6). The retrieved *TcNPRs*-like sequences were further analyzed for the identification of the presence of BTB domain and ankyrin repeats using an online database of NCBI CDD (Conserved Domain Database) (http://www.ncbi.nlm.nih.gov/Structure/cdd/wrpsb.cgi).

## Evolutionary analysis

MEGA, an offline software, was used to execute the evolutionary study of *TcNPRs* with *NPR* genes in other species. The muscle alignment technique was used to align the amino acid sequences of *TcNPR1* proteins from *T. cacao, Arabidopsis thaliana*, *Zea mays*, *Brachypodium distachyon*, *Gossypium Hirsutum*, *Malus domestica*, *Cucumis sativus,* and *Oryza sativa*. The phylogenetic tree was constructed by these protein sequences using the Neighbor-joining (NJ) algorithm in MEGA software, with a bootstrap method by several 1000 replications. The created tree was presented via the Newick file on the iTOL website (https://itol.embl.de/upload.cgi).

## Physio-chemical characteristics and subcellular localization analysis of the TcNPR genes

The Expasy program (https://web.expasy.org/protparam/) was used to identify multiple parameters, such as protein length, protein molecular weight, protein isoelectric point (pI), protein GRAVY value, and instability index, of the *TcNPR1* peptide sequences. The phytosome v13 (https://phytozome-next.jgi.doe.gov/) database was used to retrieve the names of genes and locations of *TcNPR1* peptide sequences. Additionally, the WoLF PSORT software (https://wolfpsort.hgc.jp/) was utilized to assess the subcellular location (SL) of the TcNPR genes.

## Motif identification, gene structure, and CREs identification

An online database, Gene Structure Display Server (GSDS) (v2.0) (http://gsds.cbi.pku.edu.cn/), was utilized to construct the gene structure intron-exon of the *TcNPR1* genes by using genomic and CDS sequences from phytozome v13 (https://phytozome-next.jgi.doe.gov/) database. The promoter region 1000 base pairs upstream of the start codon were extracted from phytozome v13 (https://phytozome-next.jgi.doe.gov/). The PlantCare database (http://bioinformatics.psb.ugent.be/webtools/plantcare/html) was utilized to locate the cis-regulatory elements (CREs) associated with these genes. Multiple Em for Motif Elicitation (MEME) suite software (http://meme.sdsc.edu/meme/website/intro.html) was used to identify 15 motifs and display the found motifs using TBtools [14].

## Chromosomal mapping and evolutionary analysis

Each gene's molecular evolutionary rate and duplication were determined using simple Ka_Ks calculation analysis, which was performed using CDS and gene pairing files in TBtools. The values obtained were further analyzed using the lambda (λ) value of *T. cacao* for the evolutionary year of this crop by $(T = Ks/2 \lambda)$ where $\lambda = 6.5 * E\text{-}9$. The chromosomal mapping was visualized using an advanced gene location tool in TBtools. It required four files produced using the Phytozome database for chromosomal length, location, and start and end position of the gene.

## Expression analysis

**Gene expression profiling of cacao varieties responses to *P. megakarya* inoculation.** Two variants of cocoa plants (SCA6 and NA32) were utilized to investigate the

response to *P. megakarya* inoculation. The data were collected from the database NCBI GEO (https://www.ncbi.nlm.nih.gov/geo/) PMID: 30739243 to explore the effect of *P. megakarya* inoculation on the expression profile of the NPR gene family. The gene expression profiles of cocoa genotypes were categorized into 16 subcategories at 0 hours, 6 hours, 24 hours, and 72 hours. Each interval had two different replications, resulting in two main varieties [2(varieties)*2(replications)*4(Intervals)] = 16. Treatments were applied simultaneously, but the collection of leaves was done at different times for RNA extraction and analysis, and specific sequences linked to accession IDs of BLAST NPR sequences were targeted. For subsequent research, the Statistics 8.1 pairwise comparison tool was utilized to enhance understanding of up/down-regulated gene expression.

## Synteny analysis

The presence and linkage among *TcNPR1* genes on their chromosomes were analyzed using advanced circos features in TBtools. Dual synteny analysis is used in TBTool to identify orthologous and paralogous genes in distinct species. Species'. Fa and.gff files of *A. thaliana, S. lycopersicum,* and *O. sativa* were retrieved from the phytozome database. One step MC ScanX program compared each retrieved species with *T. cacao.* To find the orthologous genes in other families, selected files (. ctl, Collinearity, and.gff) and the phytozome IDs of retrieved genes of *TcNPR1.*

## Gene Ontology (GO) and miRNA analysis

A gene ontology (GO) word enrichment investigation was conducted to further evaluate the activities of the TcNPR1 genes using GO annotations. The ShinyGo v0.745 (http://bioinformatics.sdstate.edu/go/) is an online program that helps us better understand the function of the *TcNPR1* genes in *T. cacao.* Using this online database, Shiny Go, biological, molecular, and cellular functions have been shown for the identification of miRNA(s) in all three genes of *TcNPRs,* using a CDS file for an online database psRNATarget (https://www.zhaolab.org/psRNATarget/) with a default parameter.

## Results

### Identification of NPR-1 genes in *T. cacao L.*

The *NPR1* gene family in *T.cacao* was identified using the BLAST-P program to retrieve NPR1 sequences from the whole peptide sequence of cocoa, available in the Cocoa Genome Hub, using the TBtool software. The analysis revealed three NPR genes in the cocoa genome, distributed across three chromosomes out of the total ten, which were named *TcNPR1, TcNPR2,* and *TcNPR3* based on their phylogenetic relationship to *Arabidopsis thaliana.* These three genes share similar structural features. *TcNPR1* is located on chromosome 9, spans 5,159 base pairs (bp), and encodes a protein of 592 amino acids with a predicted molecular weight of 65,375.53 Daltons (Da). This protein has an isoelectric point (pI) of 5.67 and a grand average of hydropathy (GRAVY) score of −0.226. It is anticipated to localize in the cytoplasm with a localization confidence score of 6.5 (Table 1). *TcNPR2*, located on chromosome 6, spans 4,817 bp and encodes a protein of 588 amino acids, with a molecular weight of 65,596.11 Da, a pI of 6.12, and a GRAVY score of −0.248. It is predicted to localize in the nucleus with a confidence score of 3.5. Lastly, *TcNPR3*, found on chromosome 1, spans 10,073 bp and encodes a protein of 584 amino acids with a molecular weight of 65,473.44 Da. This structural and functional characterization of the NPR1 gene family provides a foundational understanding of their roles in *T. cacao*, with implications for future studies on their involvement in plant immunity and stress responses.

## Phylogenetic analysis of NPR-1 gene.

The phylogenetic analysis included 28 NPR1 peptide sequences from various species such as *Oryza sativa L.*, *Arabidopsis thaliana*, *Brachypodium distachyon*, *Carica papaya*, *Gossypium hirsutum*, *Sorghum bicolor*, *Zea mays*, *Solanum lycopersicum*, *Solanum tuberosum*, *Fragaria vesca*, *Glycine max*, and others. The grouping of the phylogenetic tree was primarily based on the presence of *TcNPR* genes alongside *Arabidopsis thaliana* sequences within the clades (Fig 1).

**Table 1. Physiochemical properties of NPR1 gene family.**

| Transcript IDs | | Gene name | Chrom. # | Location | | Strand | Size (AA) | | Mol. W | Number of Amino Acids | pi | Gravy | No. of intron & exons |
|---|---|---|---|---|---|---|---|---|---|---|---|---|---|
| Tbtool ID | Phytozome ID | | | Start location | End location | | mRNA (CDS) | Protein length | | | | | |
| Tc09v2_p007500.1 | Thecc.09G086200 | TcNPR1 | 9 | 4527355 | 4532513 | forward | 1776 | 592 | 65375.53 | 591 | 5.67 | −0.226 | 3:04 |
| Tc06v2_p011320.1 | Thecc.06G137100 | TcNPR2 | 6 | 22399684 | 22404760 | reverse | 1764 | 588 | 65596.11 | 587 | 6.12 | −.248 | 3:04 |
| Tc01v2_p014100.1 | Thecc.01G167100 | TcNPR3 | 1 | 10499166 | 10509288 | forward | 1752 | 584 | 65473.44 | 583 | 5.77 | −0.392 | 3:04 |

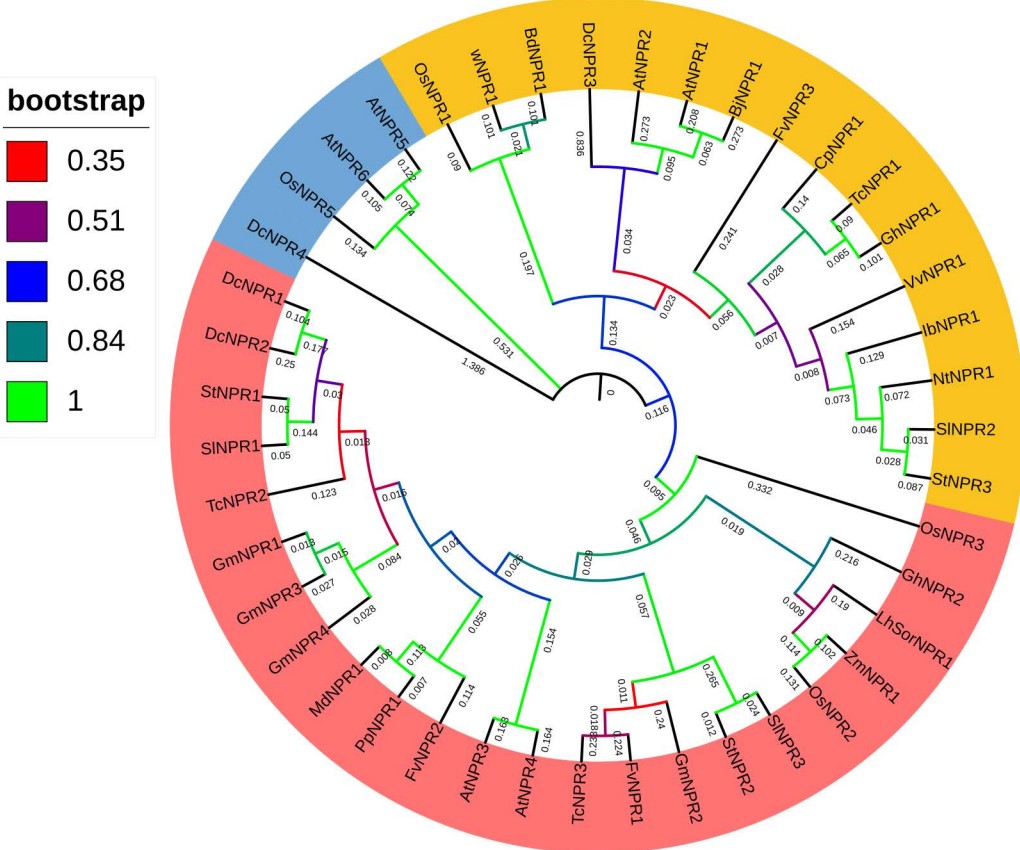

**Fig 1. Phylogenetic tree of NPR-1 genes from distinct species.** Three main clades are highlighted in different colors, and each clade is categorized based on the presence of NPR-1 of Arabidopsis and T. cacao. The species of clade III do not have any T. cacao NPR-1 gene in its evolutionary background. The color represents the bootstrap values, and the number on the inside tree represents branch length. Oryza sativa (Os), Arabidopsis thaliana (At), Zea mays (Zm), Theobroma cacao (Tc), Glycine max (Gm), Brachypodium distachyon (Bd), Gossypium hirsutum (Gh), Malus domestica (Md), Cucumis sativus (Cs).

The analysis revealed that the 44 identified proteins were categorized into three distinct subfamilies: Clade I, II, and III. The distribution of Arabidopsis NPR genes informed this clade division. Clade I contained two *TcNPR* genes, while Clade II comprised one *TcNPR* gene. In contrast, Clade III did not include *TcNPR* genes but encompassed two *Arabidopsis* NPR genes (Fig 1). These findings provide valuable insights into the evolutionary relationships and structural diversification of NPR1 genes across different species.

## Sub cellular localization

The analysis of *NPR* genes in *T. cacao* revealed distinct subcellular localization patterns for the *TcNPR* genes (Fig 2). Among these, *TcNPR3* exhibited the highest localization in the nucleus, indicating a prevalent nuclear presence, while *TcNPR1* was primarily localized in the cytoplasm, with a minor detection in the nucleus. However, *TcNPR2* did not show strong localization in any subcellular compartment. Instead, it exhibited low occurrence levels across multiple locations, including the chloroplast, nucleus, and endoplasmic reticulum (Fig 2). These findings emphasize the variability in subcellular localization of *TcNPR* genes, suggesting potential differences in their functional roles within the subcellular compartments (S1 File).

## Gene structure analysis

**Intron-exons analysis.** Next, we analyzed intron-exon structures and conserved motifs to gain more detailed insights into gene structure, domain organization, and motif composition (Fig 3). The results revealed that all three genes consistently have four exons and three introns. Notably, *TcNPR3* was found to have a particularly long intron sequence located between two of its exons (Fig 3). These findings provide valuable information on the structural organization of the *TcNPR* genes, which may have implications for their functional roles.

**Motif analysis.** The *NPR1* gene requires two main structural domains for proper functioning: BTB/BZip (Broad-Complex, Tramtrack, and Bric a brac) motif and ankyrin

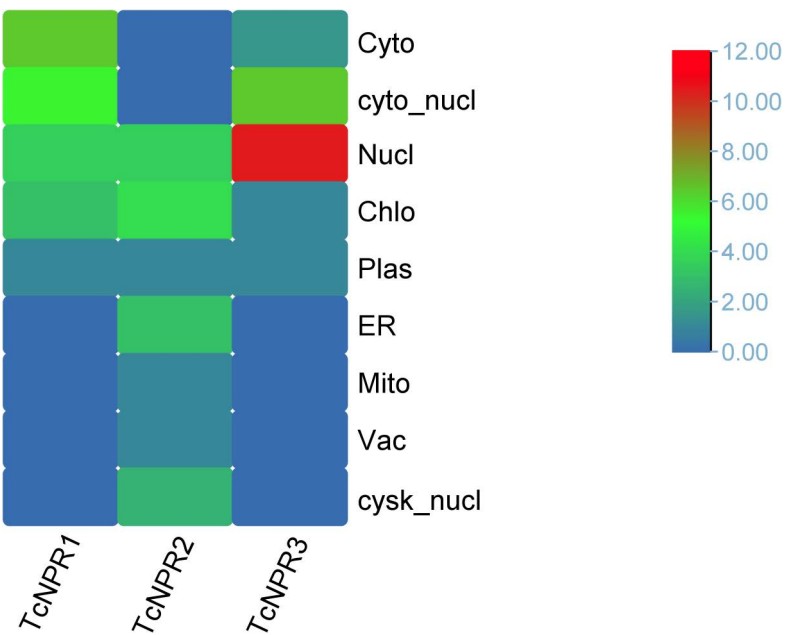

**Fig 2. Sub-cellular localization of three NPR genes across various cellular compartments.**

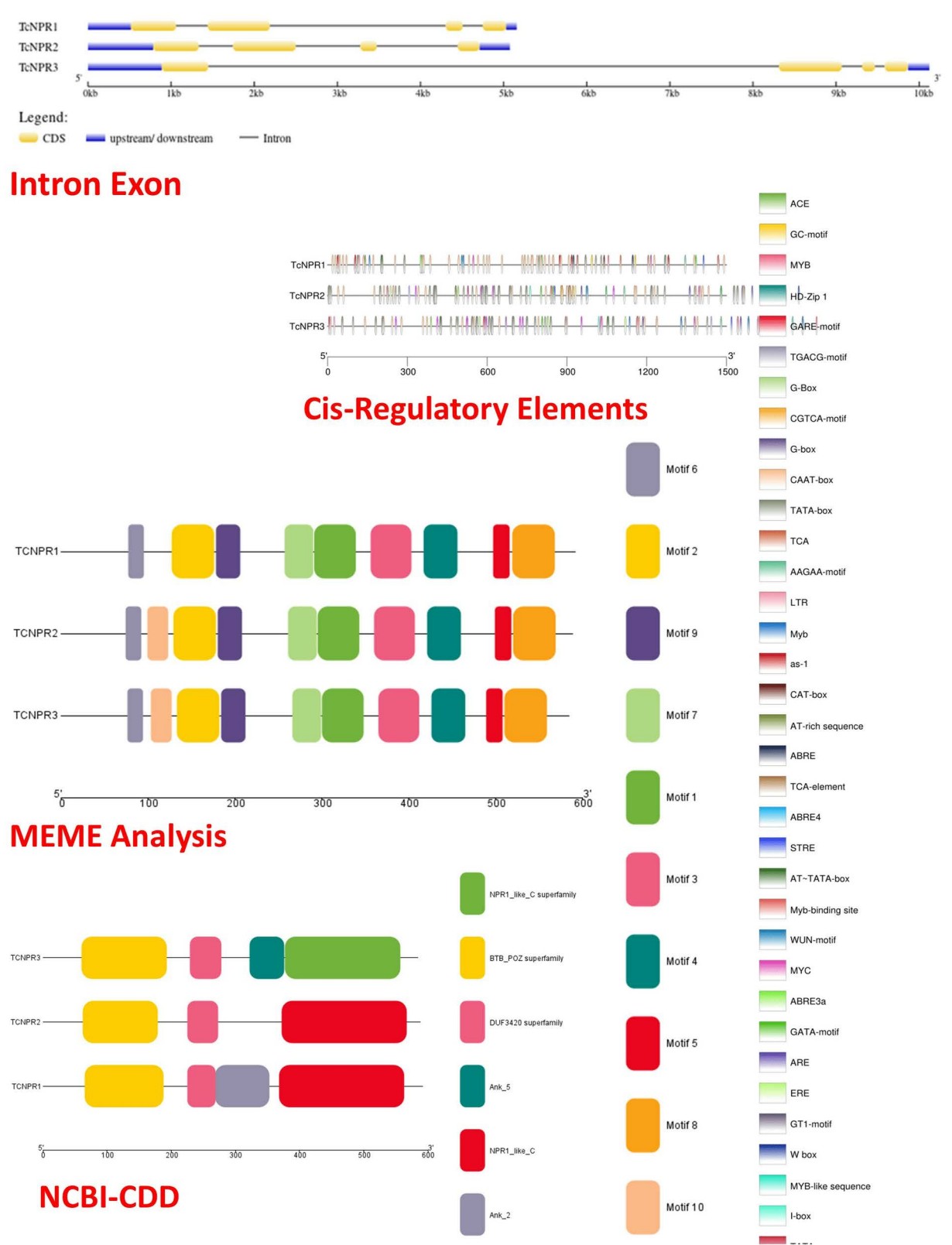

**Fig 3. Gene structural and motif analysis in three *TcNPR* genes.**

repeats. To evaluate the presence of these motifs, the MEME method in TBtools was applied to analyze the protein sequences of all 10 motifs found in the TcNPR genes. The BTB/BZip motif and ankyrin repeats were identified in all three *TcNPR* genes. Interestingly, motif 10 was absent in *TcNPR1*, while the other *TcNPR* genes exhibited all the motifs characteristic of the NPR-like superfamily (Fig 3). These motifs are important in initiating gene transcription, highlighting their essential function in regulating NPR1 gene activity.

**Cis regulatory elements (CREs).** Specific cis-regulatory elements regulate plant growth, development, phytohormone, and stress response [15]. For example, the TGACG motif influences genes involved in defensive responses and is associated with salicylic acid signaling [16]. The TATA box, a key component in the promoter region of various genes, initiates transcription, while the CAAT box enhances transcription efficiency. The auxin-responsive element (ARE) regulates genes related to development and growth. Salicylic and jasmonic acids mediate defense mechanisms, with As-1 playing a role in this process. ABA-responsive elements, including ABRE and ABRE2, regulate gene expression in response to environmental stress [17]. The interplay between the ERE and ethylene controls fruit ripening and aging.

The TCA element is key for salicylic acid-mediated defense against infection [18]. The GARE motif and TGA element regulate gibberellin-related growth genes [19]. Repetitions of TC-rich sequences are required for various hormonal responses, including those related to salicylic acid signaling, auxin-mediated growth, and WUN-motif. A comprehensive cis-regulatory analysis of NPR genes revealed 30 cis-elements associated with light response, 20 with gibberellin responsiveness, four with abscisic acid responsiveness, seven with stress response, seven MYB binding sites linked to drought inducibility, seven with anaerobic induction, and one related to salicylic acid response. The remaining elements were associated with low-temperature response, endosperm expression, and general cis-regulation, highlighting the diverse roles these elements play in regulating major physiological processes such as plant development, light responsiveness, circadian regulation, and stress responses. *TcNPR2* and *TcNPR3* were enriched in the TATA box and CAAT box motif, while *TcNPR1* has a relatively high presence of CAAT (Fig 4 and S2 File).

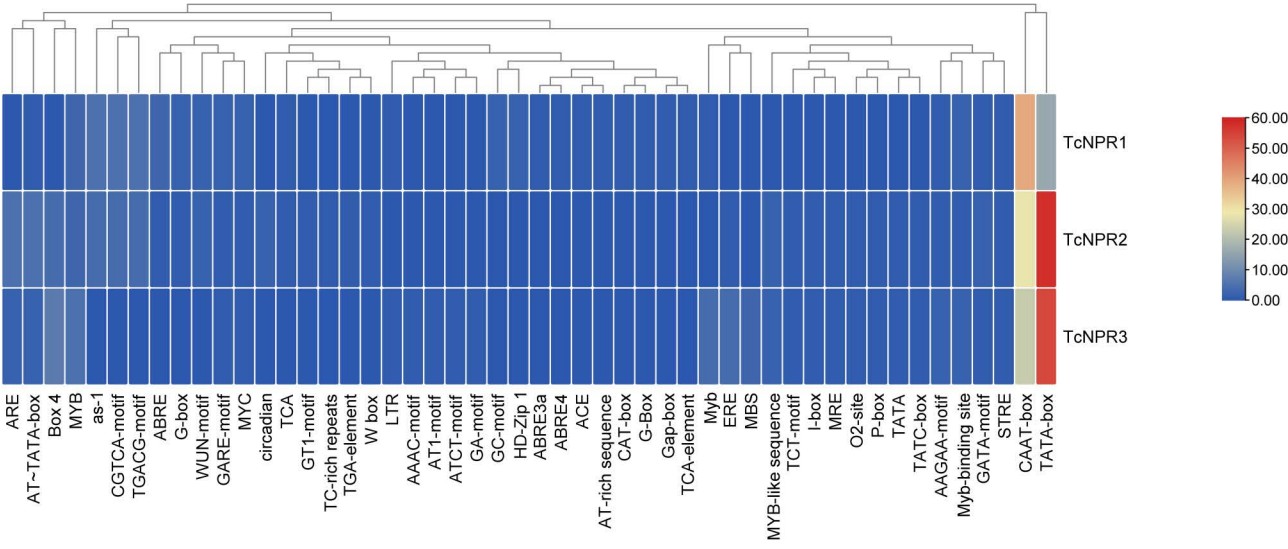

**Fig 4. Cis-regulatory elements (CREs) in various NPR genes in T. cacao.**

**Chromosomal location and gene duplication.** Chromosomal mapping reveals the location of all TcNPR genes in the cocoa plant genome. *TcNPR1* is located on chromosome 9, *TcNPR2* on chromosome 6, and *TcNPR3* on chromosome 1, indicating that these three genes are situated on different chromosomes rather than two genes on a single chromosome (Fig 5). The Ks value reflects synonymous substitutions, while the Ka value represents nonsynonymous substitutions, with the Ka/Ks ratio used as a key criterion to assess the evolutionary pressure on genes by comparing the change in amino acid sequences. The gene pairs *TcNPR1-TcNPR2* and *TcNPR1-TcNPR3* show no detectable Ks values, whereas the gene pair *TcNPR2-TcNPR3* exhibits both Ka and Ks values. The Ka/Ks ratio for this pair is 0.14335, which was used in the formula $T = Ks/(2\lambda)$ to estimate the evolutionary divergence time of these genes. The T values suggest that the *TcNPR2-TcNPR3* gene pair likely diverged approximately 160.69 million years ago (Table 2).

## Single and dual Synteny analysis

Synteny analysis was conducted on the whole genome of *T. cacao* to investigate the duplication of *TcNPR* genes on chromosomes. The study revealed no singleton genes in *T. cacao*; no duplications were detected. *TcNPR3* was identified as a paralog of *TcNPR1*, and *TcNPR2* was a paralog of *TcNPR3* (Fig 6). Furthermore, the identification of orthologous genes of *T. cacao* NPR1 in other species enabled the creation of a comparative dual synteny analysis of *T. cacao*, *Arabidopsis thaliana*, *Solanum lycopersicum*, and *Oryza sativa* (Fig 6). This dual synteny analysis identified three orthologous gene pairs with *Arabidopsis thaliana*, three with *Solanum lycopersicum*, and only one with *Oryza sativa*. The highest number of orthologous gene pairs was observed between *Arabidopsis thaliana* and *Solanum lycopersicum*, indicating a stronger

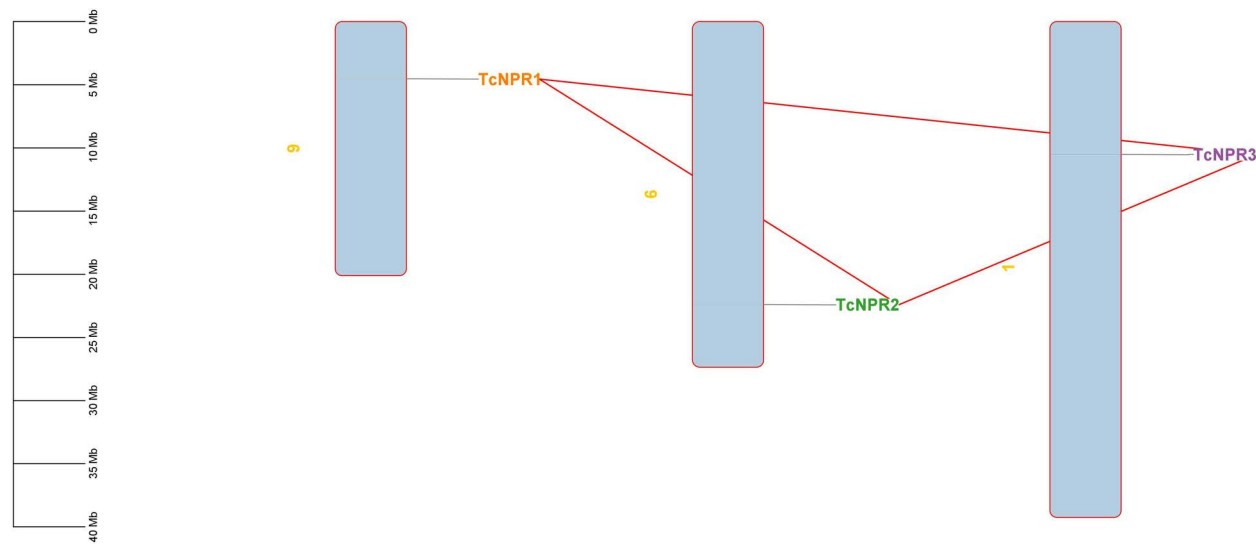

**Fig 5. Chromosomal mapping.**

**Table 2. Ka_Ks values.**

| Seq1_2 | Ka | Ks | Ka_Ks | MYA |
|---|---|---|---|---|
| TCNPR2 _TCNPR3 | 0.299479 | 2.089004 | 0.14336 | 160.6926 |

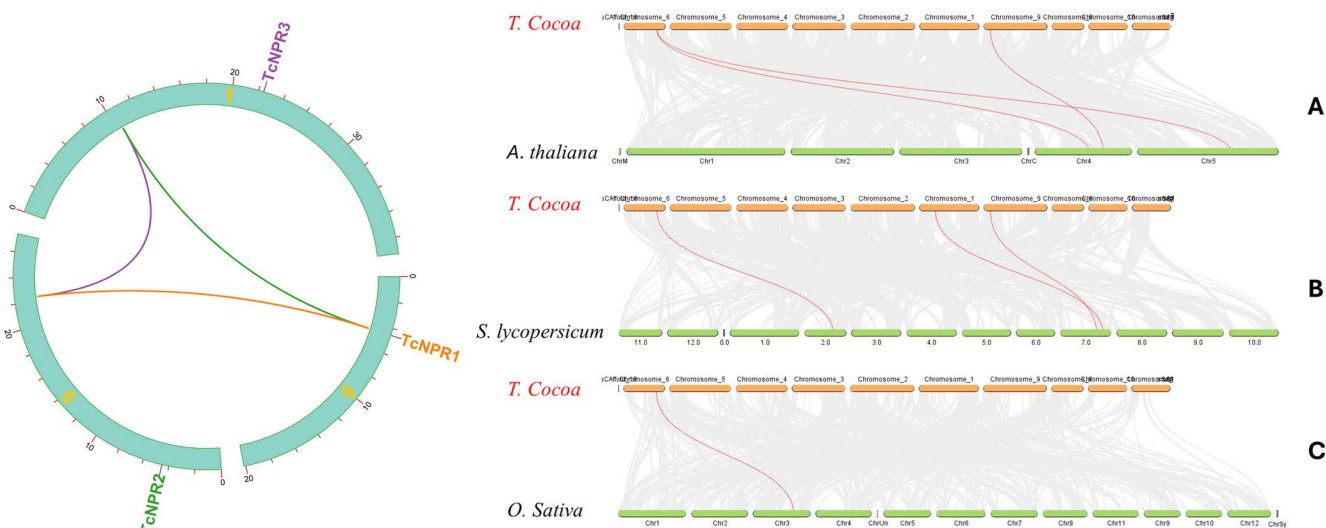

**Fig 6. Singal synteny analysis shows the presence of genes on different chromosomes (left).** Dual synteny analysis shows the presence of genes on different chromosomes (right).

evolutionary connection than *Oryza sativa*. An advanced Circos plot further illustrates that the three genes are dispersed across three chromosomes in *T. cacao*, with duplications present in *Arabidopsis* and tomato. These results highlight the evolutionary relationship of *TcNPR* genes, revealing paralogs within the genome and stronger orthologs connection with *A. thaliana* and *S. lycopersicum* compared to *O. sativa,* suggesting lineage-specific duplication.

## Gene expression analysis

For gene expression profiling, transcriptomic data were retrieved from previously published studies on *T. cacao*. This data was generated through high-throughput gene expression sequencing following treating *P. megakarya* zoospores and distilled water, sprayed onto four-month-old, grafted *T. cacao* plants (scions SCA6 and NA32). To quantify the transcriptomic response, leaf tissue was collected over a period, with RNA extracted from two leaves of separate plants serving as biological replicates. The data collection was performed at five time intervals, ranging from 0 to 72 hours. Four biological replicates were collected for each genotype and treatment at each time point, except at the 48-hour mark, when samples were only obtained from plants treated with *P. megakarya* (Fig 7). RNA extraction was performed using Invitrogen's plant RNA extraction kit. The results show a high expression of *TcNPR2* after 24 hours of both control and infection treatments, with expression levels increasing throughout the infection treatment. Both *TcNPR1* and *TcNPR2* exhibited a modest expression increase. *TcNPR2* was highly variable among the other and showed a peak expression profile at S24C, N24P, and N72P (Fig 7).

## Gene ontology (GO)

We used gene ontology (GO) categories to evaluate the TcNPR proteins using biological and molecular functional GO. In the biological process category, three significant functions related to the defense mechanisms of *T. cacao* were identified: systemic acquired resistance (GO:0009862), regulation of the jasmonic acid-mediated signaling pathway (GO:2000022), and regulation of the salicylic acid-mediated signaling pathway (GO:2000031). Most GO-term

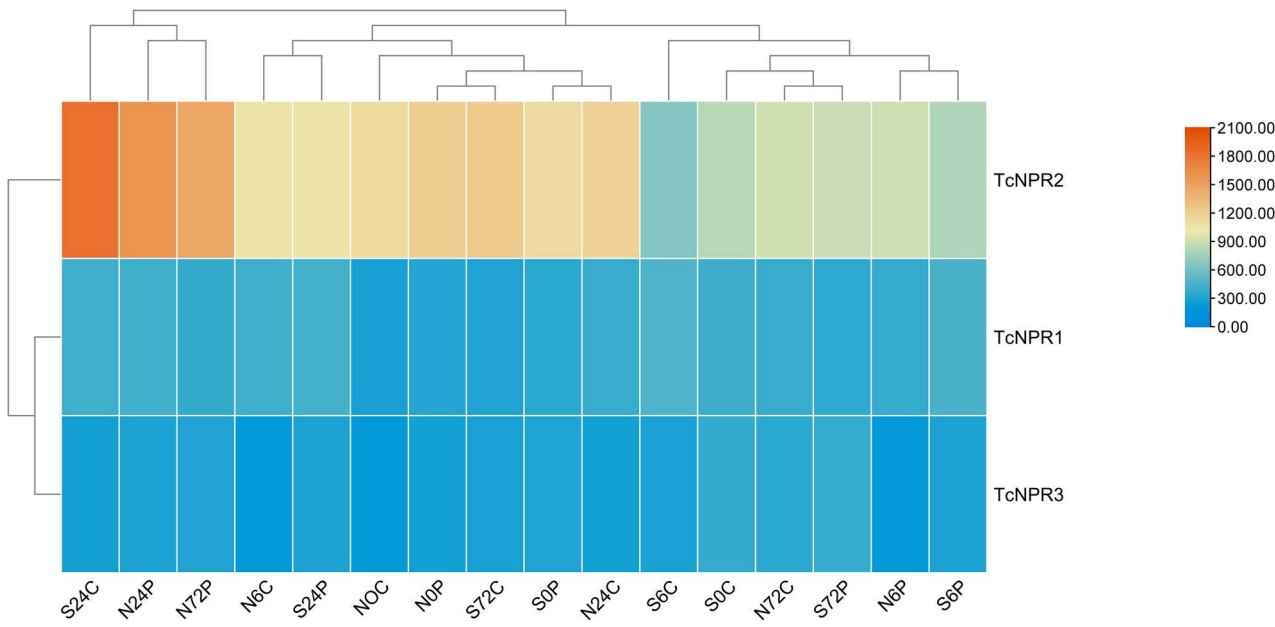

**Fig 7. Gene expression analysis of TcNPR genes retrieved from the T. cacao genome.**

categories were related to the defense response to various pathogens. These data illustrate the active response of the plant to pathogen entry, highlighting the activation of its defense mechanisms through these pathways upon pathogen infection (Fig 8).

## miRNA analysis

We analyzed the sequences targeting TcNPR genes obtained from psRNATarget, an online web server. Four miRNA sequences were identified, targeting only the TcNPR3 gene. The expected range of the target site's expectation value varied from 3.5 to 5, with miRNA lengths ranging from a minimum of 21 nucleotides to a maximum of 22 nucleotides. No miRNA sequences were found targeting *TcNPR1* or *TcNPR2*. The roles of the identified miRNAs were further analyzed using reference data from previous studies (S3 File).

## Discussion

Using a bioinformatics approach, we identified three *NPR1-like* genes, *TcNPR1, TcNPR2*, and *TcNPR3*, within the *T. cacao* genome. All the TcNPR proteins are hydrophilic, with negative GRAVY values, indicating a preference for interaction with water [20]. These are located on three different chromosomes (9, 6, and 1), which may reflect functional diversification or redundancy. Further analysis of their physiochemical properties indicated similarities among the *TcNPRs*, suggesting similar subcellular localization and protein-protein interaction potentials. The cytoplasmic and nuclear predictions for *TcNPR1* and *TcNPR3*, respectively, agreed with their functions in SA signaling and transcriptional regulation; however, the unclear localization of *TcNPR2* calls for further investigation. Phylogenetic analysis from *Oryza sativa*, *Arabidopsis thaliana*, *Brachypodium distachyon*, *Carica papaya*, *Gossypium hirsutum*, *Sorghum bicolor*, *Zea mays*, *Solanum lycopersicum*, *Solanum tuberosum*, *Fragaria vesca*, and *Glycine max* displayed that *TcNPR* genes clustered into three distinct clades. Without Arabidopsis NPR homologs, clade 1 contains *TcNPR1* clustering with Arabidopsis NPR1

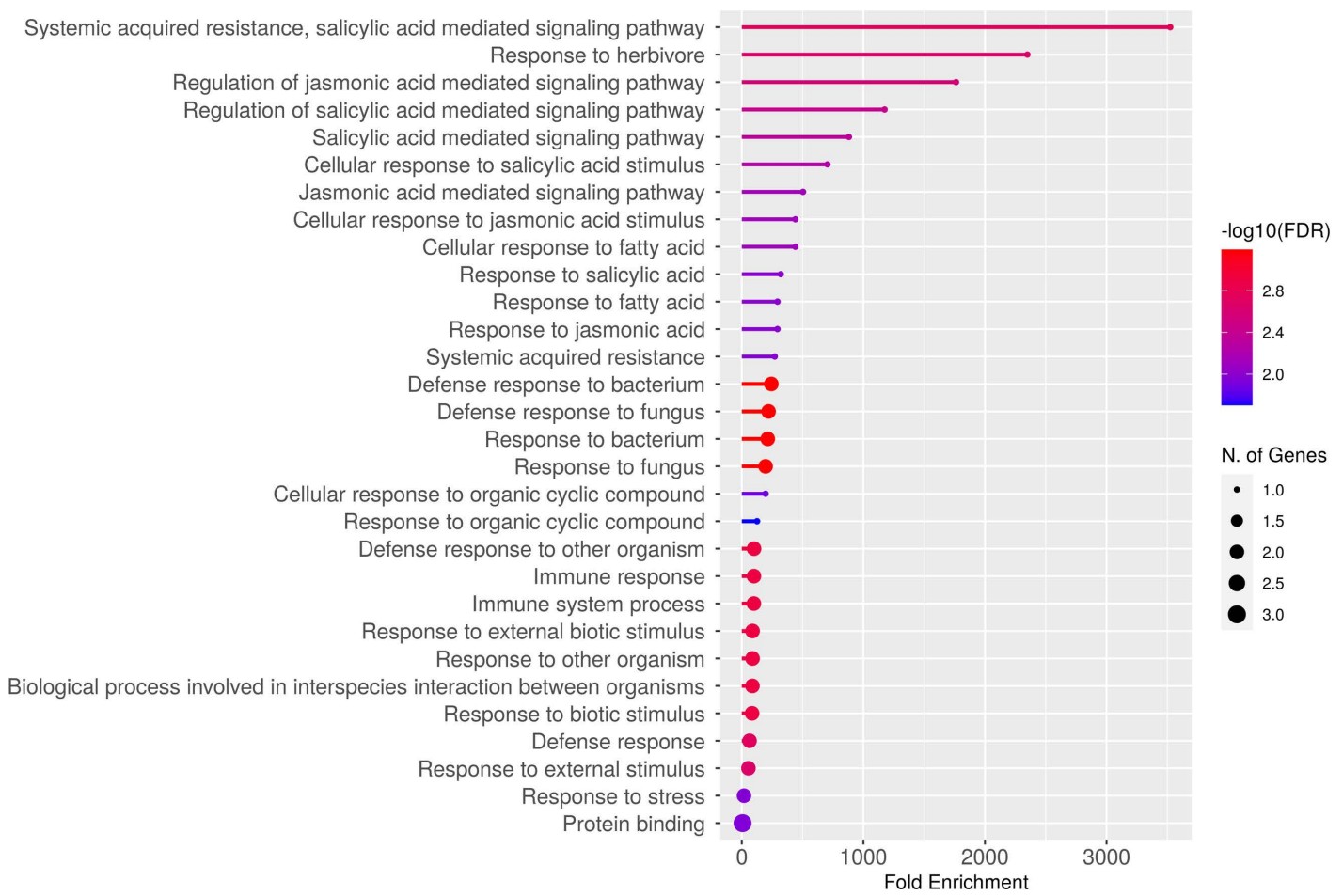

**Fig 8. Gene ontology analysis shows the role of TcNPRs in producing salicylic acid against pathogens.** The dot size represents the number of genes enriched in each category, while the heat map indicates the -log10 expression value with lowest to highest in blue to red, respectively.

and *TcNPR2* clustering with Arabidopsis *NPR2*, suggesting its conserved function in the SA signaling pathway.

In contrast, *TcNPR3* clustered into a separate clade, indicating that it may have some unique functions or an evolutionary divergence. The phylogenetic relationship showed that *TcNPR1* is the closest to *AtNPR1* and *AtNPR2*, while *TcNPR2* and *TcNPR3* are closest to *AtNPR3* and *AtNPR4*, respectively. These results indicated that the *TcNPR* genes could perform similar functions to their homologs in Arabidopsis. The GO annotations further support these observations; TcNPR genes are expressed similarly to orthologues of *Arabidopsis thaliana* [1,20]. Analysis for cis-regulatory elements shows that elements responsive to light, plant development, circadian control, proliferative gene expression, and stress response are present in *TcNPR* genes, suggesting involvement in the control of various physiological processes (Dong, 2004). These results indicate that *TcNPR* genes play a decisive role in regulating plant defense mechanisms, and environmental signals rigidly control their expression.

The evolutionary history of *TcNPR* genes can be deciphered through comparative genomic analysis across species. Our study identified three pairs of paralogous *TcNPR* genes formed by duplication events involving both tandem and segmental duplication processes. Gene

duplication has been related to evolutionary changes necessary for functional diversification. The Ka/Ks ratio of the *TcNPR2–TcNPR3* gene pair indicated divergence around 160 million years ago. Furthermore, dual synteny analysis with *Arabidopsis*, *Solanum lycopersicum*, and *Oryza sativa* verified that gene duplication occurred across species. The synteny block analysis showed that chromosome 6 has two and chromosome 9 has one *TcNPR* gene, duplicated in *Arabidopsis* and *Solanum lycopersicum*.

In contrast, only one was found on chromosome 6 in *Oryza sativa*. Using psRNATarget, four miRNAs were identified that target *TcNPR3* alone. These include miR477a, miR477b, and miR7068a, downregulating *TcNPR3* expression at post-transcriptional levels through cleavage or translation suppression. The miRNA477 family members have been reportedly involved in several plant biological processes (PMID: 38332251). These miRNAs underline the complex regulation of *TcNPR* gene expressions in response to various developmental and environmental signals. Structurally, the *TcNPR* genes shared similarities in the number of exons and introns; *TcNPR3* had a larger region than the other two. According to motif analysis, *TcNPR1* contains the same number of motifs as *AtNPR1* in *Arabidopsis*; it, therefore, may duplicate the function of *AtNPR1*. *TcNPR1* also shared similar motifs with *GhNPR1*, while *TcNPR3* showed structural diversity compared to *FvNPR3*. Although *TcNPR2* lacks an ankyrin domain, *TcNPR1* and *TcNPR3* contain conserved BTB and repeat domains critical for protein-protein interactions and transcriptional regulation.

In summary, *TcNPR* genes from *T. cacao* exhibit significant evolutionary conservation with *Arabidopsis* orthologs at the phylogenetic, functional annotation, and regulation levels. These genes participate in essential processes in the defense signaling pathways of plants against biotic stresses. Furthermore, studies on their regulation, expression patterns, and interacting microRNAs will increase understanding of their functions in response to such stresses and involvement with other physiological processes in *T. cacao*. These findings provide sound grounds for further research into the functional roles of *TcNPR* genes in this crop and their potential application in enhancing plant resilience.

## Conclusion

The NPR1 protein in *Arabidopsis* plays a pivotal role in activating the plant's immune system, particularly in response to pathogen attacks. This activation involves the synthesis of SA and establishing SAR, which helps protect the plant from subsequent infections. In this study, we identified three *NPR1-like* genes in *T. cacao* (*TcNPR1*, *TcNPR2*, and *TcNPR3*), which could play an essential role in defending against *P. megakarya*, a pathogen responsible for significant cacao diseases. Phylogenetic and functional analyses revealed that these *TcNPR* genes share similarities with their *Arabidopsis* counterparts, particularly in their involvement in SA signaling and transcriptional regulation. Notably, miRNAs targeting *TcNPR3* were identified, suggesting the potential for post-transcriptional regulation of these genes in response to *P. megakarya* infection. This points to a possible mechanism of fine-tuned regulation, wherein miRNAs modulate the expression of defense-related genes to optimize the plant's immune response. The discovery of *TcNPR* genes and their similarities with *Arabidopsis* NPR1 underscores the potential importance of these genes in regulating cacao's immune system. Identifying miRNA-targeted *TcNPR3* raises intriguing questions about the role of post-transcriptional regulation in the cacao defense response to pathogens. Given the infection-prone nature of cocoa plants, NPR1-like gene activation via SA provides a substantial strategy for designing more resistant varieties. In addition, determining the precise mechanisms by which miRNAs influence *TcNPR* gene expression during pathogen attack would further strengthen the knowledge in transgenic cocoa studies.

Future research should also explore the potential for engineering *TcNPR* genes or their regulators to enhance resistance to *Phytophthora* and other pathogens, thereby improving cacao crop resilience. Given the economic importance of cacao and its susceptibility to diseases like those caused by *P. megakarya*, advancing our understanding of the molecular mechanisms underlying pathogen resistance in *T. cacao* will be critical for sustainable crop management. Additionally, the role of *TcNPR* genes in broader plant defense mechanisms, including responses to abiotic stress and environmental factors, warrants further investigation. Ultimately, the knowledge gained from this study could contribute to developing new biotechnological tools for improving disease resistance in cacao and other crops.

## Supporting information

**S1 File. Wolfpsort score of subcellular localization.**
(XLSX)

**S2 File. CREs across three TcNPR genes.**
(XLSX)

**S3 File. miRNA enrichment across three TcNPR genes.**
(XLSX)

## Author contributions

**Conceptualization:** Muhammad Umar Rasheed, Muhammad Zeshan Haider, Adnan Sami, Muhammad Shafiq, Ansar Ali.

**Data curation:** Muhammad Umar Rasheed, Aiman Malik, Muhammad Zeshan Haider, Adnan Sami.

**Formal analysis:** Aiman Malik.

**Investigation:** Muhammad Umar Rasheed, Aiman Malik, Adnan Sami, Ansar Ali.

**Methodology:** Aiman Malik, Muhammad Zeshan Haider, Ansar Ali.

**Project administration:** Qurban Ali, Muhammad Arshad Javed, Ansar Ali.

**Resources:** Ansar Ali.

**Software:** Muhammad Umar Rasheed, Muhammad Zeshan Haider, Adnan Sami, Ansar Ali.

**Supervision:** Muhammad Shafiq, Qurban Ali, Ansar Ali.

**Visualization:** Muhammad Zeshan Haider, Adnan Sami, Ansar Ali.

**Writing – original draft:** Muhammad Umar Rasheed, Aiman Malik, Muhammad Zeshan Haider, Adnan Sami.

**Writing – review & editing:** Muhammad Shafiq, Qurban Ali, Muhammad Arshad Javed, Ansar Ali.

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
