## [Decision Letter · Decision Letter 0]

20 Dec 2024

PONE-D-24-52977NPR1-Like Genes in Theobroma cacao: Evolutionary Insights and Potential in Enhancing Resistance to Phytophthora megakarya

Dear Dr. Ali,

Thank you for submitting your manuscript to PLOS ONE. After careful consideration, we feel that it has merit but does not fully meet PLOS ONE’s publication criteria as it currently stands. Therefore, we invite you to submit a revised version of the manuscript that addresses the points raised during the review process.

You will find the comments from the reviewers below which will help you improve your manuscript.

We look forward to receiving your revised manuscript.

Kind regards,

Valentine Otang Ntui, Ph.D

Academic Editor

PLOS ONE

2. We note that your Data Availability Statement is currently as follows: [If the data are all contained within the manuscript and/or Supporting Information files, enter the following: All relevant data are within the manuscript and its Supporting Information files.] Please confirm at this time whether or not your submission contains all raw data required to replicate the results of your study. Authors must share the “minimal data set” for their submission. PLOS defines the minimal data set to consist of the data required to replicate all study findings reported in the article, as well as related metadata and methods (https://journals.plos.org/plosone/s/data-availability#loc-minimal-data-set-definition).

Additional Editor Comments:

Having gone through the manuscript, I see a major flaw with the gene expression analysis presented in Fig. 8. The authors should inoculate cacao with Phytophthora megakarya and check the expression of the mentioned genes rather than relying on information deposited on NCBI. It should be noted that several parameters would have changed from the time the information was deposited on NCBI to the time the analysis was done.

Reviewers' comments:

Reviewer's Responses to Questions

**Comments to the Author**

1. Is the manuscript technically sound, and do the data support the conclusions?

Reviewer #1: Yes

Reviewer #2: Yes

2. Has the statistical analysis been performed appropriately and rigorously? 

Reviewer #1: N/A

Reviewer #2: N/A

3. Have the authors made all data underlying the findings in their manuscript fully available?

Reviewer #1: Yes

Reviewer #2: Yes

4. Is the manuscript presented in an intelligible fashion and written in standard English?

Reviewer #1: No

Reviewer #2: No

5. Review Comments to the Author

Reviewer #1: 1. The manuscript needs major English language revision. A number of typo and grammatical mistakes have been seen in your manuscript, Please try to improve the readability of your manuscript as possible by consulting with an English language expert.

2. Abstract should be more precise. In Abstract, briefly explain how study was conducted and what measured.

3. Figure 1, The values of relationship and similarity index is missing. Try to improve the figure quality and please indicate the species names in the figure legend.

4. One limitation of this study is the reliance on the previous published data for gene expression profiling of Cacao varieties responses to Phytophthora Megakarya inoculation and the absence of experimental validation.

Reviewer #2: GENERAL

The authors used in silico approach to identify NPR-like genes in Cocoa and analyzed the implications especially in relation to their possible role in enhancing resistance to Phytophthora megakarya. No laboratory experiment was carried out.

I found that the given the nature of the study, the appropriate softwares were used for analysis and the results presented were largely reflective of the software output. There was no "big" need for the traditional statistical analysis except in Fig. 8 where I have commented below that the authors should state exactly what analysis was done since they derived raw data from a previously published paper.

Other section-by-section comments are compiled below:

INTRODUCTION

1. Overall, the manuscript is REPLETE with poor grammatical constructions that have substantially impeded logical flow and understanding of the work presented. A few examples are found in lines 49, 53, 68-69, 88, 90-91, 107-110, etc. The paper MUST NOT be accepted in the present form without a careful and extensive English language check and thorough revision. Also, attention should be given to standard scientific formats such as italicizing scientific names, capital letters beginning genus names and small letters beginning species name, writing the full form of an abbreviation at first mention (lines 70, 88, etc).

2. There are too many old references in the introduction and some of them do not seem to connect with the statements preceding them. For example, in lines 59-61, 78-80, the authors made mention of expressions like "recent studies", "until now", but cited references as old as "2004" and "2009". It would suggest that adequate literature review was not done to build in new references and connect them smoothly with the older references. As it stands currently, the older references in the manuscript seem to have been merely imported from a previous similar paper.

3. Lines 78-87: It is quite confusing whether the authors were reviewing a past literature or talking about the current study under consideration since they kept using "we/our". The entire construction needs review.

MATERIALS AND METHODS

1. Section title should be: "Materials and Methods", not "Material and Methods"

2. Be consistent with T. cacao, not T. cocoa or T. Cacao

3. TcNPR-like sequences and not TcNPR like sequences (line 123-124).

4. BTB domain means what?...line 124

5. Line 130-131: Any special reason for the choice of species whose NPR amino acid sequences were used in the phylogenetic analysis with TcNPRs?

6. Line 135: Do you mean Physico-chemical?

7. Lines 139-140: ...locations of chromosomal what???

8. Line 141: "...was used to assess..." does not sound right. Recast.

9. Line 143: CERs or CREs? At first mention, state full meaning.

10. Lines 150-152: Statement does not make a lot of sense. Review and re-present.

11. Lines 163-164: I think the sentence can be re-presented like this: "To investigate the expression levels of T. cacao NPR genes in response to Phytophthora megakarya inoculation, 2 variants of cocoa plants (SCA6 and NA32) were utilized".

RESULTS

1. Lines 221-223: Poor construction. Recast.

2. Lines 223-224: what does "thrice" mean in the context of its use?

3. Lines 278-279: The statement: "In T. cacao no gene happened to be a singleton as no duplication was found via synteny analysis" is somewhat confusing. If there was no duplication, does it not suggest that the genes exist as singletons?? Please clarify!

4. Line 290: It is unclear what analysis the authors exactly did under expression profiling. From the presentation, the transcriptomics data was obtained from a previously published report. So, what exactly did the authors do with this data to generate Fig. 8?

DISCUSSION

1. There are too many poor constructions, bad punctuations, and illogical flow. Please review extensively.

FURTHER CONSIDERATIONS

1. The authors should relook at this section. It would appear they are merely stating what they should have done instead of leaving it up to the readers to supply those information. In fact, as it stands currently, it seems as though the points raised under this section were for the authors to fill in before sending out the manuscript but somehow forgot to attend to them. This section should be re-done.

6. PLOS authors have the option to publish the peer review history of their article (what does this mean? ). If published, this will include your full peer review and any attached files.

**Do you want your identity to be public for this peer review?** For information about this choice, including consent withdrawal, please see our Privacy Policy .

Reviewer #1: No

Reviewer #2: No

---

## [Author Response · Author response to Decision Letter 1]

2 Jan 2025

5. Review Comments to the Author

Reviewer #1: 1. The manuscript needs major English language revision. A number of typo and grammatical mistakes have been seen in your manuscript, Please try to improve the readability of your manuscript as possible by consulting with an English language expert.

Response: Thank you for the reviewer for comprehensively reviewing our manuscript, the grammatical errors and language have been updated in the new version.

2. Abstract should be more precise. In Abstract, briefly explain how study was conducted and what measured.

Response: The abstract has been revised and updated.

3. Figure 1: The values of relationship and similarity index are missing. Try to improve the figure quality and please indicate the species names in the figure legend.

Response: These values of bootstrap and figure legend is updated.

4. One limitation of this study is the reliance on the previous published data for gene expression profiling of Cacao varieties responses to Phytophthora Megakarya inoculation and the absence of experimental validation.

Response: We sincerely thank the reviewer for their valuable comments; however, due to limited resources, our study was unable to do the experimental work.

Reviewer #2: GENERAL

The authors used in silico approach to identify NPR-like genes in Cocoa and analyzed the implications especially in relation to their possible role in enhancing resistance to Phytophthora megakarya. No laboratory experiment was carried out.

I found that the given the nature of the study, the appropriate softwares were used for analysis and the results presented were largely reflective of the software output. There was no "big" need for the traditional statistical analysis except in Fig. 8 where I have commented below that the authors should state exactly what analysis was done since they derived raw data from a previously published paper.

Other section-by-section comments are compiled below:

INTRODUCTION

1. Overall, the manuscript is REPLETE with poor grammatical constructions that have substantially impeded logical flow and understanding of the work presented. A few examples are found in lines 49, 53, 68-69, 88, 90-91, 107-110, etc. The paper MUST NOT be accepted in the present form without a careful and extensive English language check and thorough revision. Also, attention should be given to standard scientific formats such as italicizing scientific names, capital letters beginning genus names and small letters beginning species name, writing the full form of an abbreviation at first mention (lines 70, 88, etc).

Author Response: These mistakes have been resolved and updated in the new version.

2. There are too many old references in the introduction and some of them do not seem to connect with the statements preceding them. For example, in lines 59-61 and 78-80, the authors made mention of expressions like "recent studies" and "until now" but cited references as old as "2004" and "2009". It would suggest that adequate literature review was not done to build in new references and connect them smoothly with the older references. Currently, the older references in the manuscript seem to have been merely imported from a previous similar paper.

Author Response: These mistakes and references have been updated and resolved in the new version.

3. Lines 78-87: It is quite confusing whether the authors were reviewing a past literature or talking about the current study under consideration since they kept using "we/our". The entire construction needs review.

Author Response: These lines have been rewritten for clarity.

MATERIALS AND METHODS

1. Section title should be: "Materials and Methods", not "Material and Methods"

Author Response: This typo has been corrected.

2. Be consistent with T. cacao, not T. cocoa or T. Cacao

Author Response: We sincerely apologize for the confusion. These mistakes have been resolved and updated in the new version. T cacao is the scientific name, while T cocoa is the processed one.

3. TcNPR-like sequences and not TcNPR like sequences (line 123-124).

Author Response: These mistakes have been resolved.

4. BTB domain means what?...line 124

Author Response: BTB is explained in the updated version.

5. Line 130-131: Any special reason for the choice of species whose NPR amino acid sequences were used in the phylogenetic analysis with TcNPRs?

Author Response: There is no specific reason; the few common species were used to compare the phylogeny.

6. Line 135: Do you mean Physico-chemical?

Author Response: It is physiochemical.

7. Lines 139-140: ...locations of chromosomal what???

Author Response: These mistakes have been resolved, and the sentence has been rewritten for clarity.

8. Line 141: "...was used to assess..." does not sound right. Recast.

Author Response: This sentence is revised.

9. Line 143: CERs or CREs? At first mention, state full meaning.

Author Response: The new version has resolved and updated these mistakes.

10. Lines 150-152: Statement does not make a lot of sense. Review and re-present.

Author Response: It has been corrected.

11. Lines 163-164: I think the sentence can be re-presented like this: "To investigate the expression levels of T. cacao NPR genes in response to Phytophthora megakarya inoculation, 2 variants of cocoa plants (SCA6 and NA32) were utilized".

Author Response: These mistakes have been resolved and updated in the new version.

RESULTS

1. Lines 221-223: Poor construction. Recast.

Author Response: These sections have been revised.

2. Lines 223-224: what does "thrice" mean in the context of its use?

Author Response: We sincerely apologize for the confusion, and it has been updated.

3. Lines 278-279: The statement: "In T. cacao no gene happened to be a singleton as no duplication was found via synteny analysis" is somewhat confusing. If there was no duplication, does it not suggest that the genes exist as singletons?? Please clarify!

Author Response: The statement has been updated and clarified.

4. Line 290: It is unclear what analysis the authors exactly did under expression profiling. From the presentation, the transcriptomics data was obtained from a previously published report. So, what exactly did the authors do with this data to generate Fig. 8?

Author Response: We tried explaining this section in our updated manuscript version.

DISCUSSION

1. There are too many poor constructions, bad punctuations, and illogical flow. Please review extensively.

Author Response: This section has been revised extensively.

FURTHER CONSIDERATIONS

1. The authors should relook at this section. It would appear they are merely stating what they should have done instead of leaving it up to the readers to supply those information. In fact, as it stands currently, it seems as though the points raised under this section were for the authors to fill in before sending out the manuscript but somehow forgot to attend to them. This section should be re-done.

Author Response: This section has been removed and revised to a new version.

---

## [Decision Letter · Decision Letter 1]

15 Jan 2025

PONE-D-24-52977R1

NPR1 -Like Genes in Theobroma cacao: Evolutionary Insights and Potential in Enhancing Resistance to Phytophthora megakarya

Dear Dr. Ali,

Thank you for submitting your revised manuscript. After careful consideration, we feel that it has merit but does not fully meet PLOS ONE’s publication criteria as it currently stands. Therefore, we invite you to submit a revised version of the manuscript that addresses the points raised during the review process.

We look forward to receiving your revised manuscript.

Kind regards,

Valentine Otang Ntui, Ph.D

Academic Editor

PLOS ONE

Journal Requirements:

Reviewers' comments:

Reviewer's Responses to Questions

**Comments to the Author**

1. If the authors have adequately addressed your comments raised in a previous round of review and you feel that this manuscript is now acceptable for publication, you may indicate that here to bypass the “Comments to the Author” section, enter your conflict of interest statement in the “Confidential to Editor” section, and submit your "Accept" recommendation.

Reviewer #1: All comments have been addressed

Reviewer #2: (No Response)

2. Is the manuscript technically sound, and do the data support the conclusions?

Reviewer #1: Yes

Reviewer #2: Yes

3. Has the statistical analysis been performed appropriately and rigorously? 

Reviewer #1: N/A

Reviewer #2: N/A

4. Have the authors made all data underlying the findings in their manuscript fully available?

Reviewer #1: Yes

Reviewer #2: Yes

5. Is the manuscript presented in an intelligible fashion and written in standard English?

Reviewer #1: Yes

Reviewer #2: Yes

6. Review Comments to the Author

Reviewer #1: I appreciate the efforts made by authors to improve the readability of the manuscript in the revised version. Now the manuscript can be accepted for publication after a minor revision.

1. In abstract, state full name of the gene NPR1 at first mention.

2. Lines 25-27, Statement does not make a lot of sense, review and re-present.

3. Lines 52-53, Poor construction, rewrite the statement.

4. Lines 64-65, Are pathogens susceptible or cocoa plants are susceptible to pathogens? Please correct the statement.

5. Conclusion section should be as concise as possible and should not be more than one or two paragraphs.

Reviewer #2: There has been a major English language revision in the manuscript and it is currently much better to understand. Also, most of the issues raised in the first review seem to have been addressed.

However, a few observations are noted below;

1. Lines 41-44: Authors referred to "Recent investigations" yet cited references as old as 1999 and 2004. This should be addressed.

2. Line 257: The last sentence: "However, only one duplication is found in..." is incomplete.

3. Physiochemical or physicochemical? Be consistent. See line 294

7. PLOS authors have the option to publish the peer review history of their article (what does this mean? ). If published, this will include your full peer review and any attached files.

**Do you want your identity to be public for this peer review?** For information about this choice, including consent withdrawal, please see our Privacy Policy .

Reviewer #1: No

Reviewer #2: No

---

## [Author Response · Author response to Decision Letter 2]

15 Jan 2025

Response to Reviewers

Review Comments to the Author

Reviewer #1: I appreciate the efforts made by the authors to improve the readability of the manuscript in the revised version. Now the manuscript can be accepted for publication after a minor revision.

1. In abstract, state the full name of the gene NPR1 at first mention.

Authors Response: We would like to thank the reviewer for their time and comprehensively improving the quality of our manuscript. The full name of NPR1 is mentioned.

2. Lines 25-27, Statement does not make a lot of sense, review and re-present.

Authors Response: The sentence has been restructured.

3. Lines 52-53, Poor construction, rewrite the statement.

Author Response: This sentence has been rewritten.

4. Lines 64-65, Are pathogens susceptible or cocoa plants are susceptible to pathogens? Please correct the statement.

Authors Response: We apologize for the mistake, it has been corrected in the new version of manuscript.

5. Conclusion section should be as concise as possible and should not be more than one or two paragraphs.

Author’s Response: The conclusion section has been trimmed and rewritten to make it more concise.

Reviewer #2: There has been a major English language revision in the manuscript and it is currently much better to understand. Also, most of the issues raised in the first review seem to have been addressed.

However, a few observations are noted below;

1. Lines 41-44: Authors referred to "Recent investigations" yet cited references as old as 1999 and 2004. This should be addressed.

Author’s Response: We thank the reviewers for pointing out the mistake; the new manuscript contains the updated references.

2. Line 257: The last sentence: "However, only one duplication is found in..." is incomplete.

Author Response: This sentence was missed, and we apologize to the reviewer; it has been replaced.

3. Physiochemical or physicochemical? Be consistent. See line 294

Author Response: The word has been changed to make it more consistent.

---

## [Editor Report · Decision Letter 2]

17 Jan 2025

NPR1 -Like Genes in Theobroma cacao: Evolutionary Insights and Potential in Enhancing Resistance to Phytophthora megakarya

PONE-D-24-52977R2

Dear Dr. Ali,

We’re pleased to inform you that your manuscript has been judged scientifically suitable for publication and will be formally accepted for publication once it meets all outstanding technical requirements.

Kind regards,

Valentine Otang Ntui, Ph.D

Academic Editor

PLOS ONE

Additional Editor Comments (optional):

None

Reviewers' comments:

None

---

## [Editor Report · Acceptance letter]

PONE-D-24-52977R2

PLOS ONE

Dear Dr. Ali,

I'm pleased to inform you that your manuscript has been deemed suitable for publication in PLOS ONE. Congratulations! Your manuscript is now being handed over to our production team.

Kind regards,

on behalf of

Dr. Valentine Otang Ntui

Academic Editor

PLOS ONE
